# The Effectiveness of High-Intensity Interval Training vs. Cardio Training for Weight Loss in Patients with Obesity: A Systematic Review

**DOI:** 10.3390/jcm14041282

**Published:** 2025-02-15

**Authors:** Sławomir Jagsz, Marcin Sikora

**Affiliations:** Institute of Sport Sciences, The Jerzy Kukuczka Academy of Physical Education, 40-065 Katowice, Poland; m.sikora@awf.katowice.pl

**Keywords:** interval effort, endurance exercise, age-specific exercise recommendations, body composition changes, exercise-induced fat reduction, aging and exercise physiology

## Abstract

**Background:** Obesity is a growing public health issue, increasing the risk of metabolic disorders and cardiovascular diseases. Physical activity is a key factor in obesity treatment; however, the effectiveness of different exercise modalities remains unclear, especially considering age-related physiological differences. High-intensity interval training (HIIT) and moderate-intensity continuous training (MICT) are two commonly recommended strategies, but their impact on fat reduction across different age groups has not been thoroughly analyzed. This study aims to determine which training modality is most effective for fat reduction in individuals with obesity, considering age as a crucial factor in exercise response. **Methods:** A systematic review was conducted, analyzing studies published between 2014 and 2024. The inclusion criteria focused on the studies comparing HIIT and MICT effects on body composition in obese individuals of different age groups. The data extraction included training protocols, fat mass reduction, and adherence levels. The primary outcomes measured changes in body fat percentage and muscle mass retention. **Results:** HIIT was found to be most effective for younger individuals (18–30 years), promoting fat oxidation and muscle retention. In middle-aged adults (31–40 years), both HIIT and MICT yielded similar benefits, with MICT preferred due to better adherence. In older adults (41–60 years), MICT provided a more sustainable strategy for fat reduction and muscle preservation. However, the variability across the studies limits definitive conclusions. **Conclusions:** Age influences the effectiveness of HIIT and MICT for obesity treatment, highlighting the need for age-specific exercise recommendations. Future studies should standardize training protocols and assess long-term metabolic adaptations to optimize physical activity guidelines.

## 1. Introduction

Obesity is a major health concern in modern society, representing a risk factor for many chronic diseases, including diabetes, heart disease, and respiratory disorders [1]. The growing prevalence of obesity necessitates the search for effective fat reduction methods that simultaneously promote overall physical and metabolic fitness. Contemporary approaches to physical training emphasize optimizing training parameters to maximize results and maintain participant motivation. A key aspect of weight loss is the alteration of body composition, which is not always directly visible through weight changes but plays an important role in improving metabolic health [2]. Increased muscle mass and an improved sarcopenic index, or muscle-to-fat ratio, can occur during the early stages of training, even though overall weight loss is not yet noticeable [3]. Properly selected training, supported by appropriate energy balance and diet composition, promotes these positive changes, resulting in an improved functional capacity of the body and the long-term maintenance of health effects [2].

In the context of optimizing training effects, quality of diet is also crucial. Reducing saturated fat intake [4] and providing high-quality protein [5] and adequate hydration [4] promote not only fat reduction but also improved metabolic adaptations and muscle regeneration. The intake of protein with a high biological value supports the maintenance and development of muscle mass, which is particularly important in obese individuals, who often have a reduced amount of lean body mass.

High-intensity interval training (HIIT) is receiving increasing attention in the scientific literature as an alternative to traditional moderate-intensity training (MICT) [6]. However, the choice of the appropriate type of training may depend on a number of factors, including the exercisers’ age, fitness level, and individual exercise tolerance. The effects of these methods on lean body mass (FFM), cardiorespiratory fitness (CRF) [7] and the ability to maintain physical activity over the long term are also important considerations [8].

This problem is extremely complicated and the physical training used must allow weight loss and not reduce motivation [9]. In addition, the training must be safe. Among the modern forms of training used for people with obesity is high-intensity interval training (HIIT). This training allows individual [10] adjustments of the intensity and volume of training [11]; however, it requires experience in planning to achieve the effect of weight loss without reducing the motivation of the person exercising [12]. Exercise is widely recognized as having a positive effect on reducing body fat and improving cardiorespiratory fitness [13,14]. Moderate-intensity, constant-intensity training (MICT) workouts—often combined with strength training—are effective in reducing body fat and improving CRF [6]. However, they require a considerable time commitment. In recent years, HIIT has emerged as a popular fitness trend because of its time-saving and effective nature [15].

HIIT aids weight loss via several mechanisms. It increases energy expenditure after exercise because the phenomenon of elevated postexercise oxygen consumption contributes to the overall energy deficit required for weight loss [16,17]. In addition, HIIT increases fat oxidation [18], promoting the breakdown of stored fat into energy and contributing to reduced overall body fat. Additionally, HIIT improves metabolic function [19] by increasing insulin sensitivity and regulating glucose levels, which is particularly beneficial for people with insulin resistance. Moreover, HIIT helps maintain lean muscle mass, which is crucial for maintaining metabolic health. Research highlights the effects of HIIT on appetite-regulating hormones, including ghrelin and peptide YY [20]. Individual responses to HIIT can vary, and its effectiveness is optimized when integrated with a holistic approach, including a balanced diet and healthy lifestyle.

Taking these aspects into account, the purpose of this article is to analyze the effectiveness of HIIT and MICT training in reducing body fat and improving health parameters in obese individuals of different age groups. This article aims to review the literature from 2014 to 2024 on the effectiveness in reducing body fat in people from different age categories. An additional/applicable aim is to identify the HIIT or MICT training algorithm that is the most effective in reducing body fat in people with obesity from different age categories.

## 2. Methods

### 2.1. Identification of Keywords, Phrases, and Literature Bases

This study uses a theoretical approach based on an analysis of the scientific literature, classified according to the system developed by Montero and León (2007) [21] and Ato, López-García, and Benavente (2013) [22]. This study can be classified as a systematic review, as it is based on a structured analysis of the available literature on the effectiveness of HIIT training compared to cardio training in the context of weight reduction in obese individuals.

To conduct a comprehensive review of the scientific literature, keywords and phrases were identified and used to create a search strategy. This strategy included searching for scientific articles published between 2014 and 2024, ensuring that they were up-to-date and in line with the latest scientific research. Reputable scientific databases were selected for analysis, such as PubMed/MEDLINE, EBSCO, Google Scholar, and Web of Science, because of their broad coverage, high reputation, and ability to access peer-reviewed scientific publications from various disciplines. Google Scholar, although not a database in the traditional sense, allows access to scholarly publications and informal papers that may contain relevant information.

The main terms included in the search strategy covered various aspects of HIIT training, cardio training, obesity, and the weight loss process: (1) high-intensity interval training (HIIT)—training characterized by short, intense periods of exertion interspersed with periods of rest or lower intensity; (2) high-intensity intermittent exercise (HIIE), Aerobic Interval Exercise, and Aerobic Interval Training—various forms of high-intensity interval training; (3) cardio—moderate-intensity aerobic exercise such as running, cycling, or swimming; (4) weight loss; (5) obesity.

The literature search was conducted in accordance with the PRISMA (Preferred Reporting Items for Systematic Reviews and Meta-Analyses) guidelines. The literature search was performed on 4 November 2024, and Boolean operators such as AND, OR, and wildcards were used for the search. Here is an example search query:

(“High-Intensity Interval Training” OR “HIIT” OR “high-intensity intermittent exercise” OR “aerobic interval training”) AND (“cardio” OR “aerobic exercise”) AND (“weight loss” OR “obesity”).

### 2.2. Inclusion and Exclusion Criteria

To ensure the reliability of the analysis and the accuracy of the data collection, only articles based on clinical or experimental studies published between 2014 and 2024 on obese people aged 18–60 were included. The studies excluded were the following: (1) those including people not involved in training or weight loss, (2) articles without access to the full text, (3) studies on other age groups, (4) studies involving caloric restriction or specific diets, (5) protocols using supramaximal interval training (SIT), (6) studies involving competitive athletes.

### 2.3. Data Extraction Process

For the data extraction, a form was created to include key information such as (1) the demographic characteristics of the participants, (2) the type of intervention (HIIT or cardio training), (3) the duration of intervention, (4) weight loss parameters (e.g., weight change, waist circumference, and body fat percentage).

The analysis of each eligible publication was conducted independently by two investigators to ensure the consistency and reliability of the data collected. Any discrepancies were resolved through discussion and consensus within the research team. In case of missing relevant data or the need for additional information, the authors of the publications were contacted. Once the extraction process was completed, further statistical analysis of the collected data was performed.

### 2.4. Publication Quality Assessment

This present work uses the Critical Appraisal Skills Program (CASP). The decision to use the methodology in this analysis stems from the need for a reliable and systematic assessment of the quality of the qualified scientific studies. CASP is a widely used methodology in the critical appraisal of the scientific literature, enabling objective analysis in terms of the reliability, relevance, and usefulness of research results [23]. The use of CASP allows for the following: (1) a structured evaluation of the research—through a set of structured checklists adapted to different types of research; (2) the minimization of bias due to the methodological analysis of each study for reliability and potential risks to the results; (3) evaluating the practical utility of the results—enabling their subsequent application in research and clinical practice. By using this methodology, it was possible to conduct a transparent and reliable analysis of the available scientific reports, which increased the credibility of the conclusions of this review.

CASP enables the critical evaluation of key aspects of a study, such as methodology, sample representativeness, data analysis, and conclusions. The evaluation assessed how the participants were selected, the sample representativeness, and the tools to analyze the data. The results of the evaluation are summarized in tabular form to facilitate a comparison of the quality of the articles.

### 2.5. Study Selection and Screening Process

Our search strategy identified 808 articles on obesity from all the included databases. After removing the duplicates and articles with unavailable full content and considering the age range, 65 articles were included. After analyzing the titles and abstracts, 22 articles qualified for the literature review. After further analysis of the full content of the articles, the articles describing the effects of the types of training studied on psychological and motivational aspects were rejected. In addition, the papers containing a combination of training and dietary recommendations, as well as when the study was not conducted on obese patients, were also rejected from the analysis. A detailed overview of the included articles is presented in Figure 1.

## 3. Results

All the qualified publications were assessed for quality using the Critical Appraisal Skills Program (CASP), and the research was conducted in accordance with the Declaration of Helsinki.

Most of them scored “very good quality” (10), and the remaining scored “good quality” (1). The characteristics of the qualified publications are shown in Table 1.

Study group:

A total of 508 people were eligible for analysis. The patients with overweight and obesity included 313 women and 195 men aged 18–60 years. These individuals were also diagnosed with concomitant diseases closely related to obesity, such as type 2 diabetes, hypertension triglyceridemia, hypercholesterolemia, or metabolic syndrome.

Considering this study’s aim, the articles eligible for analysis were categorized by age: 18–30, 31–40, and 41–60 years. Additionally, two papers examining the 20–50 age range were included as a separate category. The details of the included papers are shown in Table 1.

In all the included articles, two types of training were used and compared: high-intensity interval training and constant-intensity continuous training (cardio). The exercises introduced during the intervention, as well as the intensity, volume, frequency, etc., differed among the studies.

In most studies, cycling was chosen as the physical activity. The training duration was 4–16 weeks, with the session depending on the HIIT training parameters. However, in cardio training, it was predetermined by the researchers. All the studies included in the analysis considered load progression along with the training duration. In addition, the recommendations for the training volume and exercise intensity used during the workouts were related to the general WHO recommendations [24]. An important strength of the analyzed work was the individualization of loads expressed as a percentage of maximal oxygen uptake. Furthermore, the analyzed papers included different HIIT training variations, including high-intensity interval training at 90% of VO2peak, all-out supramaximal-level sprint interval training, and supramaximal sprint interval training at 120% of VO2peak. In addition, moderate-intensity cardio-type training was used in variations, including moderate-intensity continuous training and moderate-to-vigorous-intensity continuous training. Unfortunately, not all the papers accurately defined the training protocol for the continuous exercise. A detailed description of the training parameters is provided in Table 2.

In the studies analyzed, clear training effects on body composition were observed. The papers were not uniform in expressing changes in body composition, and this effect was not considered crucial in all the articles. Nevertheless, considerable changes in body composition were observed in most of the papers analyzed after both types of training were simultaneously applied with no statistically significant differences in the body composition change results between the types of training used. The exact effects of the training provided are shown in Table 3.

## 4. Discussion

Training recommendations for individuals with obesity include starting with low-to moderate-intensity workouts to avoid the risk of injury and excessive stress on joints [36]. Intensity should be gradually increased with fitness. For training volume, three–four training sessions are initially recommended per week, lasting 30–60 min each [37]. Gradually increasing the training volume is advisable with improving endurance. Adapting training intensity and volume to individual abilities and goals is key to training effectiveness and avoiding injury [38]. The literature focuses on the link between excessive fat mass and the occurrence of metabolic diseases. Scientific reports indicate that obesity is an important risk factor for the development of cardiovascular disease, type 2 diabetes, lipid disorders, and hypertension [39,40]. Further, increased body fat can lead to inflammation and insulin resistance, thereby predisposing individuals to serious health complications [41].

This study attempts to assess which training protocol used in people with overdeveloped body fat most effectively reduces it. Recent scientific reports often compare the hitherto recommended and widely used low-intensity continuous training method—cardio—with the training often used by athletes, i.e., HIIT, which is increasingly being used among obese individuals, owing to its potential for reducing body weight and improving metabolic capacity [42].

However, some principles should be considered when using HIIT with individuals with overweight and obesity. First, the intensity and duration of the intervals should be adjusted to the individual’s capabilities and health status. People with obesity have limited physical capacity and a higher risk of injury; therefore, gradually increasing the intensity and length of the intervals is important.

In addition to individual capacity, sex-based differences in the physiological response to training should also be considered. Women, particularly those in the premenopausal and postmenopausal [43] stages, may experience different rates of fat oxidation and muscle retention due to variations in hormonal levels, particularly estrogen. Estrogen plays a crucial role in muscle preservation and lipid metabolism, which may explain why premenopausal women tend to maintain lean mass more effectively than postmenopausal women, who experience a decline in muscle mass and metabolic rates [43]. Similarly, in men, declining testosterone levels associated with andropause can impact muscle synthesis and fat distribution, potentially influencing the effectiveness of training protocols [44]. These physiological factors should be taken into account when designing exercise programs tailored to different age and sex groups.

Second, it is important to include various exercises and adapt them to the participant’s preferences and physical capabilities. HIIT training can include several exercises, such as running, cycling, skipping, or fitness machines. Monitoring the appropriate load and rest between intervals is also important. Obese people may need a longer recovery time between series; therefore, there should be adequate rest time. Scientific studies have confirmed the effectiveness of HIIT training in weight reduction in obese people. For example, the work of Alahmadi (2014) [42] found that HIIT training can significantly reduce fat. In addition, a systematic review by Wewege (2017) [6] confirmed that HIIT training can be an effective method of fat reduction in individuals with overweight and obesity, especially when compared to moderate-intensity training. Thus, using HIIT training in individuals with obesity can effectively reduce weight; however, the right principles must be followed and tailored to the individual needs and abilities of the participant.

Although individualizing the dosage of training loads seems warranted, the literature lacks specific recommendations for safe load ranges for obese individuals, particularly with regard to age. There are also no guidelines that reliably ensure effective weight reduction across different age groups.

Age is crucial for planning the training loads for individuals with obesity because of differences in physical abilities and activity levels and the risk of age-related health complications. With advancing age, changes in muscle, joint, and cardiovascular functions can occur, thereby affecting the ability to perform certain exercises and exercise tolerance [45]. For middle-aged people, considering individual limitations and health needs when planning training to avoid injury and deterioration is important [46].

Research supports the need to tailor training to the age and health status of obese individuals. For example, Miller et al. [47] emphasize the need to individualize training programs for the elderly because of differences in physical abilities and the risk of health complications. In addition, the elderly may need a longer recovery time after training and attention to selecting the intensity and type of exercise performed [48].

A discrepancy is noted in the analyzed reports, which may be related to the differences in the methods used to measure changes in body composition and the characteristics of the studied groups. The studies conducted in the youngest group primarily focused on women, whereas the people >30 years of age included those who were overweight or had obesity and those with concomitant diseases such as type 1 and type 2 diabetes and metabolic syndrome. This heterogeneity in the studies is understandable given the duration of excess body fat and its impact on the development of metabolic diseases.

Most of the analyzed papers showed decreasing body weight or fat mass with interval and cardio training [25,26,27,28,30,32,33]. Moreover, the differences observed in these parameters between the groups using different types of training were not statistically significant.

However, the rate and magnitude of body composition changes induced by different types of training may be influenced by several key variables, including training duration, intensity, and frequency. High-intensity interval training (HIIT) is known for its efficiency in reducing fat mass in shorter periods due to its impact on postexercise oxygen consumption (EPOC), whereas moderate-intensity continuous training (MICT) typically requires a longer duration to achieve similar effects. Furthermore, muscle mass development appears to be a significant factor in weight management, particularly in HIIT protocols where increased lean body mass may counterbalance weight reduction, as observed in the study by Tucker et al. (2021) [35]. The impact of exercise interventions is also age-dependent, with younger individuals often demonstrating greater muscle hypertrophy and metabolic adaptation to HIIT compared to older adults, who may require a more progressive approach to training intensity and volume.

This could suggest an arbitrariness in using different types of training, as they are equally effective in reducing excess body fat.

However, the analyzed work indicates that HIIT training is effective in developing muscle tissue, of which the high weight does not reduce body weight but only changes its composition. Another advantage is the short duration of a single training unit in comparison to moderate-intensity training. Notably, in a group of younger people—18–30 years old—HIIT and cardio resulted in a reduction in body fat and body weight [25,26,27,28]. Among middle-aged and older people, the primary effects of HIIT concerned an increase in muscle content, even with an increase in body weight, or a change in body composition, with an increase in muscle content [32,35]. Nevertheless, Kong et al. [26] observed significant changes in body composition after moderate-intensity training with reductions in total body mass (TBM, −1.8%, *p* = 0.034), fat mass (FM, −4.7%, *p* = 0.002), and percentage body fat (PBF, −2.9%, *p* = 0.016). There were no statistical changes in these body composition values in the HIIT group. Furthermore, the lack of any change in body composition in young adult women after continuous training is notable [27]. The greatest weight loss in young adults was observed after moderate-intensity training [28], which is most likely related to the lack of increase in muscle mass observed following high-intensity training. However, such a change is of little benefit from a metabolic viewpoint. Moreover, in young adults with type 2 diabetes [29], moderate-intensity training resulted in a significant reduction in body weight and BMI values. In contrast, HIIT training had no effect on these variables, which was explained by the short duration of HIIT training and limited energy expenditure. Nevertheless, no changes in body composition were observed herein; therefore, it can only be surmised that introducing HIIT training may have increased muscle mass with a concomitant reduction in body fat, which could not be observed by body mass measurements alone.

The rapid fat reduction effect in young adults observed in this review may be because of differences in metabolism, hormone levels, and body composition. In addition, the body’s adaptation to training may be important. In younger obese individuals, HIIT training may effectively reduce body weight and improve body composition. The scientific literature indicates that younger individuals may be more responsive to HIIT training for burning body fat and increasing muscle mass, because of higher metabolism levels and hormones such as testosterone [49,50]. In middle-aged and older obese people, HIIT training may be beneficial; however, the responses may be more variable. With age, metabolic processes may slow down, which may affect the rates of fat loss and muscle gain [51,52].

Moderate-intensity training had an age-independent effect on body fat reduction in the studies analyzed. This effect aligns with the scientific reports stating that in younger obese people, moderate-intensity or continuous training can effectively reduce excess body fat, especially when combined with appropriate dietary control. The scientific literature indicates that regular moderate-intensity physical activity can gradually lead to weight loss and reduced body fat in the long term [42,53,54,55]. In middle-aged and older adults with obesity, moderate-intensity or continuous training may be a safer alternative than HIIT, particularly because of the potential risk of injury or excessive stress on the joints. Research suggests that regular moderate-intensity physical activity may have benefits in reducing body weight and improving body composition, while reducing the risk of serious injury [56,57].

This analysis suggests that different training modalities, such as HIIT and MICT, had varying effects on obese individuals, particularly concerning fat mass reduction and the potential risk of health disorders. For example, the reports of Wewege (2017) and Vale (2020) [6,58] show that HIIT and MICT can significantly reduce body weight and body fat in individuals with obesity. However, risks are associated with certain types of training [59], which shows the potential for an increase in intra-abdominal fat after HIIT training, particularly in people with metabolic syndrome. Several factors may contribute to an increase in intra-abdominal fat after HIIT training. The training intensity can increase oxidative stress and inflammation, which can stimulate fat accumulation in the abdominal region [25,60]. In addition, an excessive training load can lead to hormonal disturbances, such as increased cortisol levels, which can affect fat accumulation in this area [60,61]. Notably, the body’s response to HIIT training can be highly individual. Some participants may experience a significant increase in intra-abdominal fat after HIIT training, whereas others may not. Other factors, such as diet and lifestyle, which can affect the final effects of the workout, should also be considered. Thus, when planning training programs for obese individuals, particular attention should be paid to the potential risks associated with the intensity of HIIT training, and the body’s response to training should be monitored. Moreover, considering other training methods and tailoring the training program to each individual’s needs and capabilities to ensure a safe and effective approach to fat reduction and improved health is important. Unfortunately, the studies reviewed did not provide detailed data on the changes in body composition after the forms of training used, making an accurate analysis impossible. Furthermore, the observed changes in body composition in obese middle-aged and older individuals may confirm that moderate-intensity training allows for better reduction effects compared to HIIT.

A common feature of the analyzed articles is that they emphasize the importance of the duration of training to observe significant changes in body weight and body composition (C. Martins et al., 2016) [30]. Aristizabal et al. (2021) [31] suggest that the training type is secondary and the duration of training is primary. Such discrepant results show the complexity of weight loss among individuals with obesity and that introducing additional training alone is insufficient. Most publications on the subject also point to dietary aspects, psychological aspects, motivational aspects, a background of other concomitant diseases, etc. Nevertheless, referring to the age of patients with obesity is important. This analysis indicates that HIIT training is likely more effective for younger individuals (18–30 years old), whereas moderate-intensity training is more suitable for middle-aged and older individuals. In addition, HIIT, despite its indication of post-workout excess oxygen consumption, does not always translate into weight or fat mass loss, which emphasizes the importance of an individual approach. Moreover, HIIT takes less time and is more time-saving than MICT. Nevertheless, the training type seemed to have different effects. Thus, most of this work indicates the need for an individualized selection of exercise type, as well as a tailored load selection and training progression to optimize the results. Furthermore, the importance of a personalized approach is emphasized.

Overall, the effectiveness of different training modalities varies not only by the type and intensity of the exercise but also by individual characteristics such as age, sex, and hormonal status. Younger individuals tend to exhibit a greater responsiveness to both HIIT and MICT, whereas older adults may require longer adaptation periods due to age-related declines in metabolic rate and muscle regenerative capacity. Furthermore, sex-related hormonal differences, particularly estrogen and testosterone fluctuations, influence fat distribution, muscle synthesis, and metabolic flexibility. Future research should further explore these variables to develop more personalized training protocols aimed at optimizing fat reduction and muscle preservation across different demographic groups.

Based on the findings of this review, the effectiveness of HIIT and MICT in reducing fat mass and improving body composition varies across different age groups. Younger individuals (18–30 years old) appear to benefit most from HIIT, as it enhances fat oxidation while simultaneously promoting lean muscle mass retention. The high-intensity nature of HIIT may be well tolerated in this population, leading to significant metabolic improvements and long-term training adherence.

For middle-aged adults (31–40 years old), both HIIT and MICT show comparable effectiveness, but the choice of modality should consider individual fitness levels and health conditions. While HIIT remains efficient for fat reduction, MICT may provide additional benefits in terms of cardiovascular health and joint safety, making it a viable alternative for those who experience musculoskeletal concerns.

In older populations (41–60 years old), MICT tends to be the most effective and sustainable option for long-term fat reduction and muscle preservation. The lower intensity of MICT minimizes injury risk while still offering significant improvements in body composition. However, incorporating elements of HIIT in a controlled manner may help counteract age-related declines in metabolic rate and muscle mass.

### Strengths and Limitations of This Study and Future Research Directions

The main limitation of this present study is the difficulty of clearly determining which of the analyzed training protocols is most effective for each age group. This problem stems from the heterogeneity of the data presented in the selected articles—many of them do not provide a precise description of the training regimens used or accurate measurements of the changes in body composition. The lack of uniform methodologies in the studies reviewed makes direct comparisons difficult and may affect the interpretation of the results.

Despite this difficulty, an important strength of this review is its consideration of the context of age as a key determinant of the effectiveness of various forms of physical activity. Unlike many previous studies, which focused mainly on the overall effects of HIIT and MICT in obese individuals, this paper highlights age-related physiological differences and their impact on exercise adaptation. This allows for more precise recommendations on the optimal forms of training according to age, making an important contribution to the literature on physical activity in the prevention and treatment of obesity.

In the context of future research, it seems crucial to seek a greater standardization of training protocols to enable more explicit comparisons between HIIT and MICT across age groups. An important direction for further analysis should also be to consider the long-term effects of both training methods, which would allow for the evaluation of their sustainability and effectiveness over the long term. In addition, future studies could analyze physiological variables in more detail, such as the hormonal mechanisms regulating metabolism and the effect of training on muscle mass retention, which could contribute to even more precise training guidelines tailored to the age and individual needs of obese people.

## 5. Conclusions

Overall, while individual factors such as genetics, lifestyle, and hormonal changes play a role in training outcomes, these findings suggest that exercise recommendations should be age-specific. HIIT is most beneficial for younger individuals; a combination of HIIT and MICT is effective for middle-aged adults; and MICT appears to be the safest and most effective approach for older populations. Future research should continue refining these age-based recommendations to further optimize training interventions for fat reduction and body composition improvements.

## Figures and Tables

**Figure 1 jcm-14-01282-f001:**
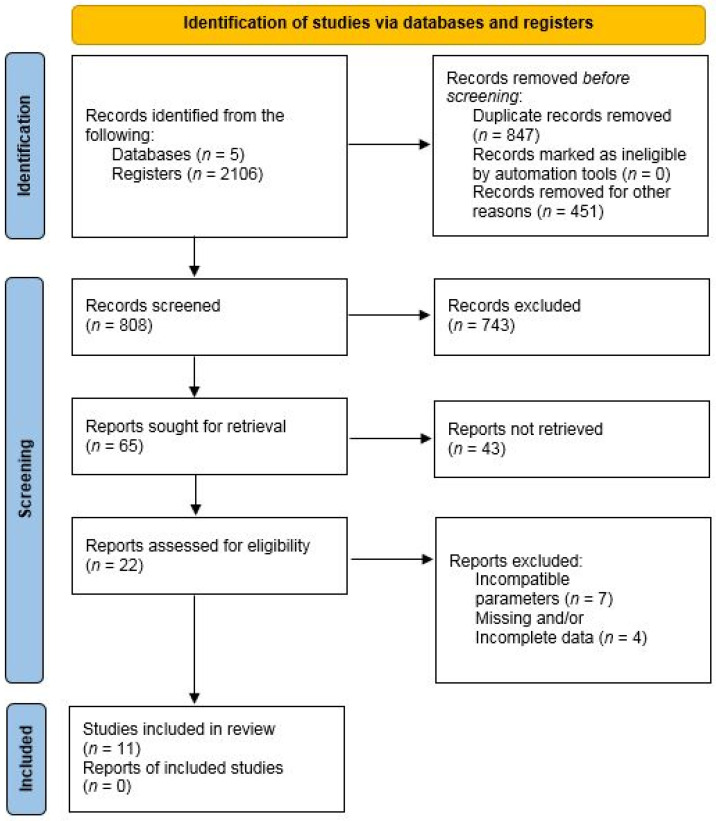
Systematic flow diagram of the review process.

**Table 1 jcm-14-01282-t001:** Characteristics of the study groups by age category.

Age Groups	Number of Publications	Number of Participants	Gender (F/M)	Number of Participants in HIIT	Number of Participants in Cardio	Group Characteristics
18–30	4	170	170/0	91	53	Overweight and obesity
31–40	2	83	30/53	45	26	Obesity and type 2 diabetes
41–60	3	203	105/98	160	43	Overweight, obesity, and metabolic syndrome
20–50	2	50	8/44	32	9	Overweight and type 1 diabetes

Abbreviations: F—female; M—male, and HIIT: high-intensity interval training.

**Table 2 jcm-14-01282-t002:** Training parameters in individual publications with age group divisions.

Study	No. of Subjects Gender	Age Group	Exercise Type	Exercise Protocol HIIT	Exercise Protocol Cardio (MICT)	Duration (Weeks)	Frequency (Per Week)
H. Zhang et al., 2017 [25]	43 F	18–30	Cycling	HIIT: 4 min cycle at 90% of VO2max and 3 min passive recovery until the target of 300 kJ; cadence: 60 rpm/min.	MICT: 60% of VO2max until the target of 300 kJ; cadence: 60 obr/min.	12	3/4
Z. Kong et al., 2016 [26]	26 F	18–30	Cycling	HIIT: 8 s sprint and 12 s passive rest for a max 60 repetitions; initial resistance was 1.0 kg, which was increased by 0.5 kg increments once an individual could complete two consecutive 20 min intermittent sprinting exercise sessions at the given intensity level.	MVCT (moderate-to-vigorous-intensity continuous exercise training): 40 min with an initial workload of 60% of VO2peak at 60 rpm/min. Once an individual had completed two consecutive exercise sessions at the specified level of exercise intensity, resistance was increased by 0.5 kg until she reached 80% of the VO2peak from the pretraining test.	5	4
H. Zhang et al., 2021 [27]	59 F	18–30	Cycling	SITall-out: 6 s sprint and 9 s passive rest—40 bouts at SIT (SIT120): a supramaximal effort of 120% of VO2peak; 1 min exercise and 1.5 min passive rest until the target of 200 kJ of work/(work rate × exercise bout duration). HIIT90: HIIT at submaximal 90% of VO2peak, 4 min exercise, and 3 min passive rest until the target of 200 kJ.	MICT: 60% of VO2peak; continuous exercise at 60% intensity until the target of 200 kJ.	12	3/4
S. Sun et al., 2019 [28]	42 F	18–30	Cycling	SIT (sprint interval training): 6 s of cycling and 9 s of passive rest for 80 repetitions (20 min); initial workload was 1.0 kg with a cadence of 100 rpm/min; the resistance was increased by 0.5 kg until arriving at 5% of the patient’s body mass.	-	12	3
J. Li et al., 2022 [29]	37 M	30–40	Cycling	HIIT: 8 min (80–95% HRmax) + 7 min (20–30% HRmax).	MICT: 30 min (50–70% of HRmax).	12	5
C. Martins et al., 2016 [30]	30 F 16 M	30–40	Cycling	HIIT: 8 s sprint and 12 s recovery phase (slow pedaling). The HIIT protocol was designed to induce a 250 kcal energy deficit, and the duration of the exercise session was individually tailored. ½ HIIT: like HIIT but only a 125 kcal energy deficit.	MICT: continuous cycling at 70% of HRmax with a 250 kcal energy deficit.	12	3
J.C. Aristizabal et al., 2021 [31]	42 F 18 M	40–60	Running	HIIT: six intervals that included 1 min of high intensity with a workload of 90% of VO2peak and 2 min with a workload of 50% of VO2peak for a total duration of 22 min.	MICT: 30 min at 60% of VO2peak for 30 min.	12	3
E. Tsz-Chun Poon et al., 2020 [32]	24 M	40–60	Running	HIIT: the HIIT group completed 10 × 1 min bouts of running at 80–90% of HRmax with a 1 min walk at 50% of HRmax in-between.	MICT: 50 min continuous jogging/brisk walking at 65–70% of HRmax.	8	3
V. Guio de Prada et al., 2019 [33]	63 F 56 M	40–60	Cycling	HIIT: 43 min of 4 × 4 min intervals at 90% of HRPEAK, interspersed with 3 min of active recovery at 70% of HRPEAK, and a cooldown period of 5 min	-	16	3
K. Minnebeck et al., 2021 [34]	8 F 16 M	Wide age range	Cycling	HIIT: 4–6 × 1 min cycling at “all-out” intensity (RPE 18–20, HR ≥ 95% HRmax) separated by 1 min of passive recovery.	-	4	2
W.J. Tucker et al., 2021 [35]	28 M	Wide age range	Cycling	HIIT: 8–11 1 min cycling intervals at 90–95% of HRmax, interspersed with 1 min active recovery periods (50 W), and a 5 min cooldown (50 W).	MICT: 30–45 min of cycling at an HR associated with 50% of VO2max and a 5 min cooldown.	4	4

Explanation of the abbreviations used in the table: HIIT—high-intensity interval training; HIIT 90—high-intensity interval training at 90% of VO2peak; ½ HIIT—high-intensity interval training with a 125 kcal deficit; MICT—moderate-intensity continuous training; MVCT (or MVICT)—moderate-to-vigorous-intensity continuous training; SIT all-out—all-out supramaximal-level sprint interval training; SIT 120—supramaximal sprint interval training at 120% of VO2peak; HRmax—maximum heart rate; HRPEAK—peak heart rate; RPE—rating perceived exertion; F—female; and M—male.

**Table 3 jcm-14-01282-t003:** Observed effects of training in all analyzed reports divided into age groups.

Study	Age Group	Observed Effect
H. Zhang et al., 2017 [25]	18–30	After 12 weeks of exercise, significant reductions in body weight and percentage body fat were observed in both intervention groups (*p* < 0.05). Changes in body weight and percentage body fat were not significantly different between the group (*p* > 0.05).
Z. Kong et al., 2016 [26]	18–30	In the MVCT group, there was a significant reduction in TBW, −1.8% (*p* = 0.034); FM, −4.7% (*p* = 0.002); and PBF, −2.9% (*p* = 0.016). There were no statistical changes in these measures of body composition in the HIIT group (*p* > 0.05). In addition, there were no significant differences between the groups in the change in values (TBW, FM, and PBF) of these variables before and after training.
H. Zhang et al., 2021 [27]	18–30	After the 12-week intervention, a reduction was observed relative to TBW, PBF, and WBF in the SIT all-out, SIT 120, and HIIT90 groups (*p* < 0.05), and the changes in each variable did not differ between the three groups (*p* > 0.05). In MICT, a similar reduction was observed only in PBF and WBF (*p* < 0.05) but not in TBW (*p* > 0.05). No variable changed under the CON (*p* > 0.05).
S. Sun et al., 2019 [28]	18–30	After 12 weeks of training, significant reductions in TBW (*p* < 0.001) and BMI (*p* < 0.001) were observed in all three groups (SIT, HIIT, and MICT). They reduced TBW by −4.9% (SIT), −5.6% (HIIT), and −6.6% (MICT). BMI decreased by 1.3 kg/m^2^ (SIT), 1.5 kg/m^2^ (HIIT), and 1.8 kg/m^2^ (MICT). However, there was no group difference in weight loss and BMI among the study groups (*p* > 0.05).
J. Li et al., 2022 [29]	30–40	That study was designed to compare relevant health indicators and not TBW or WBF (T2DM). Nevertheless, there were statistical differences in TBW and BMI in the MICT group, whereas only BMI was statistically different in the HIIT group, potentially because of the short duration of HIIT training and limited energy expenditure.
C. Martins et al., 2016 [30]	30–40	No significant changes were observed in the TBW or WBF parameters. The importance of exercise duration as a key factor affecting fat or weight reduction was emphasized.
J.C. Aristizabal et al., 2021 [31]	40–60	MICT and HIIT similarly (~1%) decreased PBF and WBF without changing TBW. FFM increased with MICT and HIIT. Therefore, HIIT was 39% more time-efficient than MICT (HIIT: 22 min; MICT: 36 min), which is explained by excess postexercise oxygen consumption and appetite regulation. HIIT and MICT could not reduce TBW more markedly, potentially because of less than recommended weekly physical activity relative to the training protocols applied. The type of exercise is possibly of secondary importance, with the duration being primary.
E. Tsz-Chun Poon et al., 2020 [32]	40–60	Both groups showed a significant percentage loss of WBF, but only the MICT group showed a significant decrease in weight, BMI, and waist circumference. There were no differences between the groups.
V. Guio de Prada et al., 2019 [33]	40–60	Only reductions in TBW and WBF were observed after the training intervention with no significant differences between the groups. HIIT resulted in similar reductions in TBW and PBF in both sexes. Thus, exercise appears to have a similar effect on improving body composition in men and women with metabolic syndrome. Similar weight reduction in both sexes showed a stronger association with an improved MetS Z-score than any of the CRF parameters analyzed. This indicates a key role for weight reduction in the prevention of metabolic syndrome.
K. Minnebeck et al., 2021 [34]	Wide age range	Patients with T1DM were studied. No changes in body composition were observed.
W.J. Tucker et al., 2021 [35]	Wide age range	TBW increased after HIIT training (+1.2 kg, *p* = 0.02) with no change in the control and MICT groups. TBW in the HIIT group increased because FFM increased (0.9 kg, *p* = 0.02). WBF did not change in any group. There were significant interindividual differences in the variability of changes in TBW (−1.8 kg to +3.1 kg), fat mass (−2.2 kg to +1.7 kg), and FFM (−1.5 kg to 2.5 kg). Visceral adipose tissue increased in the HIIT group by 6.4% (~88 cm^3^, *p* = 0.04) with no changes observed in the control and MICT groups.

Explanation of the abbreviations used in the table: *p*—probability; MVCT—moderate-to-vigorous-intensity continuous training; MICT—moderate-intensity continuous training; TBW—total body weight; HIIT—high-intensity interval training; CON—control group; FM—fat mass; PBF—percentage body fat; WBF—whole body fat; SIT all-out—all-out supramaximal-level sprint interval training; SIT 120—supramaximal sprint interval training at 120% of VO2peak; HIIT 90—high-intensity interval training at 90% of VO2peak; BMI—body mass index; T2DM—type 2 diabetes mellitus; MetS—metabolic syndrome; CRF—cardiorespiratory fitness; T1DM—type 1 diabetes mellitus; and FFM—fat-free body mass.

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
