# Peer review of "The Effectiveness of High-Intensity Interval Training vs. Cardio Training for Weight Loss in Patients with Obesity: A Systematic Review"

_jcm, 2025, doi:10.3390/jcm14041282_

Round 1

Reviewer 1 Report

Comments and Suggestions for Authors

The article “Effectiveness of High-Intensity Interval Training vs. Cardio 2 Training for Weight Loss of Patients with Obesity: A Scientific  Review

Abstract: Modify the abstract

Introduction

Modify the introduction by adding new references.

Last paragraph should be rewritten

Methods

Methods should be mentioned with references

MINOR COMMENTS

Various situations should be considered that will increase the research value.

Grammatical Errors should be removed. Typos should be corrected.

Comments on the Quality of English Language

Grammatical Errors should be removed.

Reviewer 2 Report

Comments and Suggestions for Authors

Effectiveness of high-intensity interval training vs. cardio training for weight loss of patients with obesity: a scientific review

Obesity treatment guidelines suggest moderate-intensity continuous training (MICT), but the patient's compliance to this indication remains low. High-intensity interval training (HIIT) is a time sparing training mode whose metabolic effects are not clear. Current review of the literature from 2014 14 to 2024 aims to find the determinants of the effectiveness of undertaking physical activity in the form of high-intensity interval training or moderate-intensity, constant-intensity training effort and, consequently, the most efficient possible reduction of body fat mass concerning the age of the exercising person. Table 1 presents  the training parameters in individual publications with age group division. A systematic review would have been more appropriate References must be up- dated

Reviewer 3 Report

Comments and Suggestions for Authors

The authors present a good research paper.
• Relevance of the topic: Excellent.
• Introduction: Good.
• Methodology: Poor.
• Results: Poor.
• Discussion: Good.
However, Reconsider after major revision. In general, the paper follows an adequate structure and correct scientific support. However, there are a series of constraints that should be considered.

Specific comments.

The whole text need major revision and try not to use the pronoun "we or they". In the same way, reviewing each section's verb tense is recommended. Results (past tense), Introduction (past tense), Method (past tense), Results (past tense), Discussion (present tense), and Conclusions (present tense). In the same line, the author must review all the quotes and reference style, it doesn’t fit the journal requirements. Also, the tables and figures must be reviewed and shown as the journal rules.

1. Title.

That is a good selection of the title. It shows the aim of the study.

2. Abstract.

2.1.) Including the heading in the abstract body is necessary. (Background, design and method, results and conclusions).
2.2.) The abstract must follow the structure mentioned above. Please, rewrite the whole abstract, it is necessary to add more information about the method followed, the results obtained and add a short sentence related to the conclusions. In the same line, the aim of the study is not clear, it is necessary to rephrase the aim sentence.

3. Keywords.

3.1.) Try not to use the same word as in the title. Please, modify the keywords.

4. Introduction.

4.1.) This section presents a coherent and clear manner with the correct support of the scientific literature.

4.2.) Lines 62-67, this section must be included in the introduction, it is not necessary to be separated with a heading.

5. Materials and Method.

5.1.) Design. The authors must include a “design” section. This section must be ubicated in the “2.1.” heading. To establish the design followed to carry out the present study (theoretical study). It’s recommended to use the following methodologist:
1. Montero, I.; León, O.G. A Guide for Naming Research Studies in Psychology. Int. J. Clin. Heal. Psychol. 2007, 7, 847–862.
2. Ato, M.; López-García, J.J.; Benavente, A. A Classification System for Research Designs in Psychology. Ann. Psychol. 2013, 29, 1038–1059, doi:10.6018/analesps.29.3.178511.

5.2.) Did you follow the PRISMA statements? When did you carry out the document search? Include the date of the search. It is recommended to include at the end of this section the final phrase used in the databases, to improve the compressive of the information, and what type of Booleans operators you used.

5.3.) “2.4. Publication Quality Assessment”. The section is well structured and developed, but if you use an instrument to assess the quality of the studies, you must include the reference of this instrument and the reason why you used it.

5.4.) Line 116. You must include the meaning of “The CASP methodology”, what is the meaning of the abbreviation? You must include it the first time you used the abbreviation.

6. Results

6.1.) Lines 123-133 must be relocated in the method section, this information is related to the procedure and document search.
6.2.) Figure 1. You have to adequate the figure to the PRISMA statements.
6.3.) The tables don’t follow the journal requirements. They must be done again.

7. Discussion

7.1.) The discussion section is well structured, and the main findings are discussed correctly.
7.2.) “Strengths” section. It is necessary to add information related to future research lines proposed by the authors and also the strengths of the study must be shown.

8. Conclusion.

8.1.) It is well structured.
8.2.) “Limitations of the Study”. This section must be included in the discussion section without the heading.

9. References.

The references are not correct. Please, follow the instructions from the journal. The references must follow the following style:
References must be numbered in order of appearance in the text (including citations in tables and legends) and listed individually at the end of the manuscript. We recommend preparing the references with a bibliography software package, such as EndNote, ReferenceManager or Zotero to avoid typing mistakes and duplicated references. Include the digital object identifier (DOI) for all references where available.
Citations and references in the Supplementary Materials are permitted provided that they also appear in the reference list here.
In the text, reference numbers should be placed in square brackets [ ] and placed before the punctuation; for example [1], [1–3] or [1,3]. For embedded citations in the text with pagination, use both parentheses and brackets to indicate the reference number and page numbers; for example [5] (p. 10), or [6] (pp. 101–105).
1. Author 1, A.B.; Author 2, C.D. Title of the article. Abbreviated Journal Name Year, Volume, page range.
2. Author 1, A.; Author 2, B. Title of the chapter. In Book Title, 2nd ed.; Editor 1, A., Editor 2, B.,
Eds.; Publisher: Publisher Location, Country, 2007; Volume 3, pp. 154–196.
3. Author 1, A.; Author 2, B. Book Title, 3rd ed.; Publisher: Publisher Location, Country, 2008; pp.
154–196.
4. Author 1, A.B.; Author 2, C. Title of Unpublished Work. Abbreviated Journal Name year, phrase
indicating stage of publication (submitted; accepted; in press).
5. Author 1, A.B. (University, City, State, Country); Author 2, C. (Institute, City, State, Country).
Personal communication, 2012.
6. Author 1, A.B.; Author 2, C.D.; Author 3, E.F. Title of Presentation. In Proceedings of the Name
of the Conference, Location of Conference, Country, Date of Conference (Day Month Year).
7. Author 1, A.B. Title of Thesis. Level of Thesis, Degree-Granting University, Location of
University, Date of Completion.
8. Title of Site. Available online: URL (accessed on Day Month Year)

Reviewer 4 Report

Comments and Suggestions for Authors

The authors of the manuscript entitled "Effectiveness of High-Intensity Interval Training vs. Cardio Training for Weight Loss of Patients with Obesity: A Scientific Review" have chosen a very interesting and topical topic.

Several aspects I believe should be addressed and included in the manuscript:

The introduction should include information related to body composition transformations such as increased muscle mass and improved sarcopenic index, the stage in which weight loss is often not visible but beneficial changes occur for the body. The quality of the diet should also be discussed, since it is very important to know that a reduction in saturated fat consumption, the consumption of quality proteins and efficient hydration help to improve transformations during physical activity.

Within the results, it is interesting to observe whether the scientific material analyzed also referred to caloric intake or caloric restrictions, dietary recommendations or nutritional supplements associated with physical exercise, because these can significantly modify the pace of transformations in the body, and if such information appears, it is good to present and specify the impact on the observed results. Table 3 should also include information related to muscle mass or the sarcopenic index.

Discussions require a broader detail related to the impact of different types of physical activities on changes in the body, centered on the rhythm of transformations correlated with their duration, intensity, frequency, muscle mass development, age and sex.

The conclusions need to be improved by including information related to factors that influence body changes in addition to physical activity, and limitations should also include other aspects, for example physiological changes during age that influence basal metabolism (premenopause, menopause, andropause), etc.

Comments on the Quality of English Language

English expression needs to be improved.

Round 2

Reviewer 3 Report

Comments and Suggestions for Authors No comments.

Reviewer 4 Report

Comments and Suggestions for Authors

Accept in present form